# Vitamin B12 Supplementation in Diabetic Neuropathy: A 1-Year, Randomized, Double-Blind, Placebo-Controlled Trial

**DOI:** 10.3390/nu13020395

**Published:** 2021-01-27

**Authors:** Triantafyllos Didangelos, Eleni Karlafti, Evangelia Kotzakioulafi, Eleni Margariti, Parthena Giannoulaki, Georgios Batanis, Solomon Tesfaye, Kοnstantinos Kantartzis

**Affiliations:** 1Diabetes Center, 1st Propaedeutic Department of Internal Medicine, Medical School, “AHEPA” Hospital, Aristotle University of Thessaloniki, 54621 Thessaloniki, Greece; linakarlafti@hotmail.com (E.K.); evelinakotzak@hotmail.com (E.K.); margareleni@yahoo.gr (E.M.); gbatanis@yahoo.gr (G.B.); 2Department of Nutrition and Dietetics, University General Hospital of Thessaloniki ‘’AHEPA’’, 54621 Thessaloniki, Greece; nenagian@yahoo.com; 3Diabetes Research Unit, Royal Hallamshire Hospital, Sheffield Teaching Hospitals NHS Foundation Trust, Sheffield S10 2JF, UK; solomon.tesfaye@nhs.net; 4Department of Internal Medicine IV, Division of Endocrinology, Diabetology and Nephrology, University of Tübingen, 72076 Tübingen, Germany; Konstantinos.Kantartzis@med.uni-tuebingen.de; 5Institute for Diabetes Research and Metabolic Diseases (IDM) of the Helmholtz Centre Munich at the University of Tübingen, 72076 Tübingen, Germany; 6German Center for Diabetes Research (DZD), 72076 Tübingen, Germany

**Keywords:** diabetic neuropathy, vitamin B12, SUDOSCAN, metformin, diabetic foot, diabetes mellitus, autonomic neuropathy, painful neuropathy

## Abstract

Aim: To investigate the effect of normalizing vitamin B12 (B12) levels with oral B12 (methylcobalamin) 1000 μg/day for one year in patients with diabetic neuropathy (DN). Patients and methods: In this prospective, double-blind, placebo-controlled trial, 90 patients with type 2 diabetes on metformin for at least four years and both peripheral and autonomic DN were randomized to an active treatment group (n = 44) receiving B12 and a control group (n = 46) receiving a placebo. All patients had B12 levels less than 400 pmol/L. Subjects underwent measurements of sural nerve conduction velocity (SNCV), sural nerve action potential (amplitude) (SNAP), and vibration perception threshold (VPT), and they performed cardiovascular autonomic reflex tests (CARTs: mean circular resultant (MCR), Valsalva test, postural index, and orthostatic hypotension). Sudomotor function was assessed with the SUDOSCAN that measures electrochemical skin conductance in hands and feet (ESCH and ESCF, respectively). We also used the Michigan Neuropathy Screening Instrument Questionnaire and Examination (MNSIQ and MNSIE, respectively) and questionnaires to evaluate quality of life (QoL) and level of pain (pain score). Results: B12 levels increased from 232.0 ± 71.8 at baseline to 776.7 ± 242.3 pmol/L at follow-up, *p* < 0.0001, in the active group but not in the control group. VPT, MNSIQ, QoL, pain score, SNCV, SNAP, and ESCF significantly improved in the active group (*p* < 0.001, *p* = 0.002, *p* < 0.0001, *p* < 0.000, *p* < 0.0001, *p* < 0.0001, and *p* = 0.014, respectively), whereas CARTS and MNSIE improved but not significantly. MCR, MNSIQ, SNCV, SNAP, and pain score significantly deteriorated in the control group (*p* = 0.025, *p* = 0.017, *p* = 0.045, *p* < 0.0001, and *p* < 0.0001, respectively). Conclusions: The treatment of patients with DN with 1 mg of oral methylcobalamin for twelve months increased plasma B12 levels and improved all neurophysiological parameters, sudomotor function, pain score, and QoL, but it did not improve CARTS and MNSIE.

## 1. Introduction

Diabetic neuropathy (DN) is one of the most common microvascular complications of diabetes. Its most prevalent forms are peripheral (DPN), autonomic (DAN, with the most commonly diagnosed form being cardiovascular autonomic neuropathy (CAN)), and painful (PDN). At the time of diagnosis of diabetes, 10–18% of patients present with nerve damage, but neuropathy has also been shown to occur even in prediabetes [1]. Peripheral/distal symmetric polyneuropathy may develop in up to 50% of patients [2], can cause sensory symptoms [3], and is associated with foot infections, ulcers, Charcot arthropathy, fractures, and amputations [2]. The prevalence of CAN is around 7% by the time of type 2 diabetes (DM2) diagnosis and increases with increasing diabetes duration by 4.6% to 6% per year [4,5], as well as to up to 65% of patients with longer durations of diabetes [6]. Though CAN is considered to be an independent predictor of cardiovascular mortality, it is often underdiagnosed [7]. One third of patients develop PDN [8] with neuropathic pain and symptoms such as burning, “pins and needles,” painful cold or hot sensations, “electric shock”-like pain, numbness and dead feeling in the feet and legs, and contact pain (allodynia) [9,10,11], and these have a major impact on the quality of life [8,12,13].

Strict glycemic control has been considered to be the cornerstone of the treatment of DN. However, it has only modest effects on the progression of DN [14], with a reduction in the progression of CAN [15], but little or no effect in DPN [15,16,17]. Nonetheless, large studies [18,19] do not agree on the optimum level of glycemic control for preventing neurophysiological deterioration. It is of particular note that strict glycemic control needs to be maintained for three-to-five years to provide any clinical benefit [20].

In addition to any antiglycemic therapy aiming at strict glycemic control, vitamin B12 (B12) has been probably the most commonly used supplement. The first reason is that vitamin B12 deficiency is quite common in patients with DM2, and the second is that vitamin B12 deficiency may cause neurological disorders, such as peripheral, autonomic (including cardiovascular), and painful neuropathy, resembling or accelerating the progression of DN [1,7]. Importantly, neuropathy from vitamin B12 deficiency can occur in the absence of the typical hematological characteristics (e.g., megaloblastic anemia and pancytopenia) of B12 deficiency [6,21]. Notably, it has been suggested that, especially in diabetic patients aged over 60 years, the cut-off for B12 levels potentially leading to neurological dysfunction should be shifted from 150 to 400 pmol/L [22]. 

According to several reports, the prevalence of vitamin B12 deficiency in DM2 may exceed 50% of patients [23,24,25,26,27,28]. The American Diabetes Association (ADA) therefore recommends that in patients with DN on metformin therapy, vitamin B12 levels should be regularly monitored on an annual basis [24,29]. At least half of patients with DM2 are older than 60 years, an age group with a prevalence of confirmed vitamin B12 deficiency varying from 12% to 23% [30]. However, vitamin B12 deficiency in DM2 has been mostly attributed to the use of metformin. Metformin-associated vitamin B12 deficiency has been known for over 40 years, and this relationship has been confirmed by several interventional studies, observational studies, and meta-analyses [21,23,24,25,26,27,30,31]. The strongest evidence was brought by a 4.3-year randomized clinical trial (RCT) by De Jager et al. [28]. From these studies, it has become apparent that metformin causes vitamin B12 deficiency in a dose- and duration of treatment-dependent manner [24], and that although it may occur as early as only four months after the initiation of metformin, vitamin B12 deficiency appears usually after at least four-to-five years of metformin use [7]. It has been reported that metformin reduces vitamin B12 uptake in terminal ileum in about 30% of patients [32]. However, the exact mechanisms by which vitamin B12 inadequacy arises in chronic-treated patients are not clear yet [6].

Based on the aforementioned data, several trials have tested the effect of vitamin B12 supplementation on DN [33,34,35,36,37,38,39]. However, in all of these studies, either vitamin B12 was given together with other supplements or glycemic control significantly improved during the follow-up, making it difficult to distinguish the effect of vitamin B12 supplementation from that of better glycemic control on DN. Consequently, it remains unclear whether DN improves purely by the normalization of B12 levels in the presence of good glycemic control. 

We therefore undertook the present study to investigate the efficacy of normalizing vitamin B12 levels with 1000 μg of methylcobalamin (C_63_H_91_CoN_13_O_14_P) daily in DM2 patients with good glycemic control and generalized neuropathy (both DPN and CAN) who were vitamin B12-deficient, partly because they had been on metformin treatment for at least four years. 

## 2. Materials and Methods

### 2.1. Patients Recruitment

The study was performed at the diabetes clinic of the University General Hospital in Thessaloniki, Greece, from January 2018 to February 2020. Ninety adult patients were enrolled (see Appendix A, Consort Flow Diagram) who fulfilled the following inclusion criteria. The first inclusion criterion was being adults (>18 years old) with DM2 and established DN, both peripheral and autonomic (DPN and DAN, respectively). The diagnosis of DN was set when two or more cardiovascular autonomic reflex tests (CARTs) were abnormal for DAN and when abnormal nerve conduction velocity levels with abnormal Michigan Neuropathy Screening Instrument Questionnaire (MNSIQ) and Michigan Neuropathy Screening Instrument Examination (MNSIE) were present for DPN [4,5,40,41]. The second inclusion criterion was having a good glycemic control (glycated hemoglobin (HbA1c) between 6.5 and 7.5%) that was stable in the last year before participating in the study. The third inclusion criterion was having metformin treatment for at least 4 years. The fourth inclusion criterion low vitamin B12 levels according to suggested normal values for DM2 patients over 60 years old (<400 pmol/L) [22]. Exclusion criteria were pernicious anemia, alcoholism, gastrectomy, gastric bypass surgery, pancreatic insufficiency, malabsorption syndromes, chronic giardiasis, acute infection or cardiovascular event in the last 6 months, surgery involving the small intestine, or HIV infection. Patients with an estimated glomerular filtration rate (e-GFR) <50 mL/min/1.73 m^2^ based on the Chronic Kidney Disease Epidemiology Collaboration (CKD-EPI) formula [42,43] or taking multivitamins or B12 supplements in the last 12 months were also excluded from the study. 

### 2.2. Randomization and Allocation

Patients were randomized to receive a new oral dispersible tablet of B12 containing 1000 μg of methylcobalamin (B12 fix 1000 mcg; UNI-PHARMA S.A.) or placebo daily for 12 months. The tablets containing B12 and the placebo had a similar appearance and were packed in similar containers. 

The allocation and randomization of participants in the two groups were performed by a computerized random sequence of numbers. The randomization order and type of treatment were also concealed from the responsible researcher and statistician. All tests and measurements were performed by a physician blind to the type of supplement that each patient received. There were no drop-outs, as all patients completed the study. The study was registered at *clinicaltrials.gov* (Identification Number NCT04706377) and approved by the Scientific Board and Bioethics committee of our University Hospital (Registration Number 42439). The study was conducted according the principles of Declaration of Helsinki. All participants signed an informed consent form. 

### 2.3. Antidiabetic and Concomitant Medication

All patients were treated with a combination of metformin and other oral antidiabetic drugs (dipeptidyl peptidase 4 (DPP4) inhibitors, glucagon-like peptide-1 (GLP-1) agonists, and sodium glucose transporters 2 (SGLT-2) inhibitors) or with a combination of metformin with basal insulin analog and other oral antidiabetic drugs (DPP4 or/and SGLT-2). All patients had been on metformin treatment for at least 4 years. No patient received a sulfonylurea or was on a basal–bolus insulin regimen. A total of 78.6% of our patients had hyperlipidemia (total cholesterol >200 mg/dL, high-density lipoprotein (HDL) cholesterol <40 mg/dL, and low-density lipoprotein (LDL) cholesterol >130 mg/dl), with 74.3% of the study population receiving statins. Patients had stable glucose control throughout the study period, and no change was needed in antiglycemic and concomitant medication. With the exception of paracetamol, which was occasionally used for 2–3 days, the patients were not allowed to take any analgesic other medication throughout the study.

### 2.4. Measurements and Tests

The assessment of sudomotor function was performed with the SUDOSCAN device, which makes use of a electrochemical reaction between sweat chloride and stainless steel electrodes. SUDOSCAN measures electrochemical skin conductance (ESC), the ratio between the current measured and the voltage applied, and expresses it in microsiemens (μS) [44,45,46]. Sural nerve conduction velocity and amplitude were measured using “DPN-check” (Neurometrix Inc., Waltham, MA, USA) [47]. The validation and details of this device are described elsewhere [47]. The vibration perception threshold was measured using a Biothesiometer (Ohio, USA) [48]. The diagnosis of DAN was done according to the Toronto Consensus panel guidelines and CARTs, which are considered the gold standard [5]. CARTs were performed with the HOKANSON ANS Reader and included R-R variation during deep breathing (mean circular resultant (MCR)), Valsalva maneuver (Vals), a 30:15 ratio expressed as a postural index (PI), and blood pressure response to standing (orthostatic hypotension (OH)) [4,5,41]. Age-specific values were applied [49]. For the clinical diagnosis of DPN, the MNSIQ and MNSIE scoring algorithm was used (result ≥2 for lower extremity examination, ≥7 for questionnaire, and ≥2.5 for MNSI examination were considered abnormal) [40,50,51]. The evaluation of quality of life status was assessed by the Diabetes Quality of Life Brief Clinical Inventory, a brief form of the questionnaire used in the Diabetes Control and Complications Trial (DCCT) trial [52]. The level of pain was evaluated with the PAINDetect questionnaire that is used for the identification of neuropathic components [53]. All measurements were performed according to international guidelines regarding patient preparation, i.e., hydration status, physical activity, and hypoglycemic events in the day before examination [4,5,51].

### 2.5. Measurement of Vitamin B12

Vitamin B12 was measured with a Cobas e 602 analyzer with electrochemiluminescence (ECLIA). Information of the technique can be found elsewhere [54,55,56]. The results are given in pmol/L. All other biochemical parameters (blood count, lipids, lipoproteins, etc.) were measured with the same analyzer. 

### 2.6. Statistical Analysis

Statistical analyses were performed with IBM SPSS v24 [57]. All continuous variables were normally distributed and are expressed as mean ± standard deviation. Differences of parameters at baseline between the active and the placebo group were tested using independent samples two-tailed *t*-test. Differences in variables between baseline and follow-up in each group were tested by a paired samples *t*-test. To compare the mean difference of the change between two groups, multiple general linear regression (ANCOVA) was used adjusted for HbA1c and antiglycemic medication. *p* < 0.05 was considered statistically significant.

## 3. Results

There were no significant differences between the active and placebo groups regarding demographics, anthropometric measurements, laboratory measurements, and neuropathy test results at baseline, as shown in Table 1, Table 2 and Table 3. The mean age of the whole patient cohort was 63 years old, and the mean duration of DM2 was 12.8 years. All patients had taken metformin for at least four years (active group: 12.9 ± 8.8 years; placebo: 10.1 ± 5.8 years). No patient was on basal–bolus insulin treatment.

Vitamin B12 levels significantly increased in the active group from 232.0 ± 71.8 pmol/L at baseline to 776.7 ± 242.3 pmol/L at follow-up, but they did not significantly change in the placebo group (from 230.9 ± 85.9 to 242.8 ± 100.7, *p* = 0.338). 

During the follow-up, the vibration perception threshold (VPT), MNSIQ, quality of life (QoL), pain score, sural nerve conduction velocity (SNCV), sural nerve action potential (amplitude) (SNAP), and electrochemical skin conductance in feet (ESCF) significantly improved in the active group (*p* < 0.001, *p* = 0.002, *p* < 0.0001, *p* < 0.000, *p* < 0.0001, *p* < 0.0001, and *p* = 0.014, respectively), whereas the indices of CARTS and MNSIE did not significantly improve (Table 4). None of the investigated parameters improved in the placebo group, and it is noteworthy that MCR, MNSIQ, SNCV, SNAP, and pain score significantly deteriorated (*p* = 0.025, *p* = 0.017, *p* = 0.045, *p* < 0.0001, and *p* < 0.0001, respectively—see Table 4). We found no significant change of blood pressure, serum lipids, and lipoproteins during the follow-up. Importantly, glycemic control was fairly acceptable at baseline and did not worsen during the 12 months of the study. 

We found a significant difference in the change from baseline to follow-up between the active and placebo groups in B12 levels (*p* < 0.001), MNSIQ (*p* < 0.001), QoL (*p* = 0.001), SNAP (*p* < 0.001), SNCV (*p* ≤ 0.001), VPT (*p* = 0.007), pain score (*p* < 0.001), and ESCF (*p* = 0.008). The change of all other indices and biochemical parameters did not significantly differ between the two groups (Table 4). 

Not a single adverse event suspected or possibly related to the new oral dispersible tablet of B12 was reported.

## 4. Discussion

DN is a serious complication of DM2, as it can cause devastating symptoms, including unremitting and unbearable pain, and can have severe and potentially life-threatening consequences, such as the so-called “diabetic foot” that is associated with ulcers, infections, Charcot arthropathy, and amputations [41,58,59,60]. In particular, CAN is a well-known independent predictor of cardiovascular mortality [7]. Currently, the only established non-symptomatic treatment of DN is strict glycemic control, which has to be maintained for several years and has a rather limited efficacy on DPN [15,16,18]. There is thus an urgent need for an effective drug therapy that would act either causally or by modifying the pathophysiology of DN. Among other drugs and supplements (e.g., alpha lipoic acid, carnitine, folate, and superoxide dismutase), vitamin B12 has perhaps been the most often used [3,33,34,35,36,38,61,62,63,64]. This is because vitamin B12 deficiency is commonly found in patients with DM2 and vitamin B12 deficiency may cause a variety of neurological disorders resembling or accelerating DN. Accordingly, several trials have investigated the effect of supplementation of B12 in DN [3,33,34,35,36,38,51,61,62,64,65]. Some of them reported favorable effects [3,33,36], but no one could prove a “pure” effect of B12 supplementation on improvements in peripheral nerve function. The majority of the studies were performed in DM2 patients with poor glycemic control (HbA1c: 8–10%) [34,36,61] and reported concurrent decrease of HbA1c levels or/and body weight during follow-up [34,36,61]. In other studies, combinations of B12 with other supplements or agents, not solely B12, were given [3,33,34,36,38,51,61,62,64,65]. In addition, there was a great variation in these studies in terms of the baseline blood levels (150–450 pmol/L), dose (25–2000 μg), duration (from 12 to 24 weeks), molecular form (cyano-, methyl-, or hydroxocobalamin) [31,35,62,63,66], and mode of administration (orally or parenterally) of B12 [31], as well as the inclusion criteria of the study participants (without or with established DN, DPN, or painful DN) [3,33,34,36,38,51,61,62,64,65]. 

Τhe present randomized controlled trial was conducted with the aim of answering at least the first of the aforementioned questions, i.e., whether B12 supplementation (alone) has an effect on DN independent of the glucose control. We therefore included patients with DM2 who had acceptable glycemic control (HbA1c levels: 6.5–7.5%) at baseline. In addition, since the good glycemic control had to be stable in the last year prior to participation in the study, the patients were expected to remain well-controlled during the follow-up. Indeed, there was no significant change of HbA1c levels during the trial period in both the active treatment and the placebo groups. Regarding the outcome of the trial, we found that the sole administration of methylcobalamin in a daily dose of 1000 μg in DM2 patients for a year exerted a beneficial effect on all indices, except for CARTs and MNSIE, of peripheral neuropathy, including neurophysiological parameters, sudomotor function, level of pain, and quality of life. This trial is thus, to our best knowledge, unique in the literature, not only because it examined the pure effect of B12 on DN in DM2 patients with good glycemic control but also because participants had all established peripheral sensorimotor, autonomic, and painful neuropathy. 

Whether the levels of B12 at baseline are critical for B12 to exert any effect on DN (or on some forms of it) is not clear from the literature. It is of particular note that it has been suggested that (especially those over 60 years old) diabetic patients may exhibit neurological dysfunction signs even if B12 levels are above what has been considered the normal level of 150 pmol/L. According to this suggestion, B12 levels of 150–400 pmol/L should be considered as a “relative” B12 deficiency in people with diabetes [22]. In several published studies, B12 levels at baseline were not measured or not mentioned [3,35,62], probably because B12 was given principally for its putative analgesic action. In other studies, baseline B12 levels were normal [33,34]. These studies generally showed little effect of B12 on DN, but the results were not uniform [33,34,35]. The present trial was designed to include patients with “relative” or absolute B12 deficiency, first because a clear increase in B12 level was required if its potential independent effect on DN was to be examined and second because there was a need to study any analgesic and other disease-modifying effects of B12 (e.g., on indexes of nerve function such as neurophysiological parameters). To enrich our sample with B12-deficient patients, we only screened those taking metformin for at least four years, which is the time period believed to be needed for the development of metformin-associated B12 deficiency [28]. According to the overall favorable effect of B12 administration on DN shown in our trial, and since almost 95% of the participants had this “relative” B12 deficiency (Table 2 and Table 4), it is reasonable to recommend B12 supplementation in every patient with DN and B12 levels below 400 pmol/L. Of note, at these levels, the classic hematological signs of B12 deficiency (megaloblastic anemia or pancytopenia) are rarely present, which was also the case in our study (Table 2). 

Similarly to the baseline levels of B12, there has been no consistency in existing trials in the literature in terms of the dose, duration, molecular form, and way of administration of B12 that are most effective on DN. Most trials had a follow-up period of 12–24 weeks [33,34,35,36,38,65], and no one exceeded 24 weeks, as was the case in the present trial, which lasted one year. In these trials, B12 had been administered in a dose varying from 250 to 2000 μg when given orally and from 1000 to 2000 μg when given intramuscularly [35,63,67,68]. For instance, doses of 250 mcg cyanocobalamin for one year [51] and 1500 mg methylcobalamin for four months [33] led to improvements in neurophysiological parameters and a substantial increase of B12 levels. However, in our previous study, 250 mcg of cyanocobalamin for one year did not result in the normalization of B12 levels [51]. Clearly, the mode of administration also plays a critical role. Traditionally, B12 is given intramuscularly in order to overcome any intrinsic factor deficiency. However, an oral tablet would have several advantages, such as patient autonomy, increased patient satisfaction, reduced treatment costs, and fewer risks for patients receiving anti-coagulants [67,68,69]. Studies investigating this possibility have suggested that oral administration has equal to intramuscular effects [66,67,68,69,70,71] ameliorating both biochemical and clinical manifestations of vitamin B12 deficiency [66,71]. Evidence points out that, out of large doses of cyanocobalamin given orally, 1% is absorbed and provides increase in serum B12 levels so that the daily requirements are covered (for review see [72]). Based on these data, we chose in our trial to assign a daily oral dose methylcobalamin of 1000 μg/24 h to the active group. This dose indeed normalized B12 levels and may accordingly be recommended, at least in patients with (absolute or “relative”) B12 deficiency. We used a new dispersible tablet preparation with an enhanced bioavailability after being dispersed in the mouth. Sublingual administration has the additional advantages of allowing for vitamin B12 absorption by sublingual capillaries [67,68], thus being well accepted by patients with swallowing difficulties and ensuring higher adherence rates. 

With respect to our findings, the improvement in SNCV was in accordance with other studies, both in humans and animals [36,39]. The findings were compatible with the reported action of methylcobalamin on the synthesis and regeneration of myelin [73,74], as well as enhanced nerve conduction and decreased neurotransmitter levels [74]. 

More importantly, we found that B12 supplementation resulted in improvements of somatosensory symptoms such as pain and paresthesia. Though our patients did not present with extremely painful neuropathy (mean pain score about 19 out of 38) [53], the pain level in the present study significantly decreased by 7%. Similar effects could be shown in our previous study [51] and other studies [3,33,63,75]. These findings once again confirm the analgesic action of B12 [63,73], possibly mediated by an increase of the availability and effectiveness of noradrenaline and 5-hydroxytryptamine [76] in the descending inhibitory pain modulation system (endogenous opioid system). 

Unexpectedly, the active treatment group did not show a significant improvement in CARTs. However, the placebo group showed a trend towards the deterioration of CARTS, suggesting a potential favorable effect of B12 on DAN despite the absence of a statistically significant result. This aspect is corroborated by a few studies that have suggested a beneficial effect of methylcobalamin on autonomic symptoms and specifically on CAN [7,35]. Whether the beneficial effect is exerted mainly in the parasympathetic [7] or sympathetic [77] nervous systems remains unknown. A particular way to estimate autonomic (dys)function is to measure sudomotor function by SUDOSCAN, a point-of-care device. Sudomotor function depends not only on the sympathetic cholinergic system but also on somatic innervation. Therefore, SUDOSCAN may be used as a simple method to screen and monitor the progression of peripheral nerve and cardiac sympathetic dysfunction in the context of a busy diabetes clinic [78]. SUDOSCAN provides excellent sensitivity and good specificity for DPN, as well as moderate sensitivity for CAN. Of particular importance is that ESCF is considered to be a more sensitive marker for detecting neuropathy compared to ESCH [44,79,80,81,82]. ESCF readings are closely related to nerve conduction scores [47,78,83]. Altogether, these data argue that SUDOSCAN may be a reliable and early detector of small fiber neuropathy [78], in addition to DPN and sudomotor dysfunction, all of which are critical factors for the development of the “diabetic foot.” It is of note that sudomotor dysfunction in DM patients is often underestimated and therefore remains undiagnosed in clinical practice [49]. In the present study, both ESCH and ESCF significantly improved in the active group, indicating a favorable effect of B12 supplementation in helping to prevent “diabetic foot,” at least in the specific group of diabetic patients. However, this potential benefit will need to be confirmed in other well-designed and large prospective studies [44,78,80,81,84,85,86]. 

There were no adverse events of the administration of B12, as other studies have also shown [3,33,34,51,61].

Our study had certain limitations. All participants were followed in a single diabetes center and were Caucasian. This ensured the homogeneity of the sample population but did not allow for the extrapolation of the results to other populations. We did not measure glutamic acid decarboxylase, intrinsic factor, or parietal cells autoantibodies because our population had confirmed DM2 and we had excluded patients with pernicious anemia. In addition, we only measured the functionality of sural nerve and not of nerves in arms and hands. Sural nerve function assessment is widely accepted to represent the respective function of other nerves too. We also did not measure homocysteine and methylmalonic acid, because they are often not available in the daily practice of the diabetes clinic, have high costs, and are not necessarily needed for the diagnosis of vitamin B12 deficiency.

In summary, this study showed that the increase of B12 levels with an oral dispersible tablet containing 1000 μg methylcobalamin for 12 months in patients with DN improved the patients’ neurophysiological parameters, sudomotor function, pain score, and QoL. Future studies will need to address still open questions, such as whether in the absence of B12 deficiency supplementation with the vitamin has any effects on established DN, whether B12 administration has favorable effects on the prevention of DN or the prevention of deterioration of subclinical DN, and which forms of DN are mostly improved by B12 supplementation.

## Figures and Tables

**Table 1 nutrients-13-00395-t001:** Demographic characteristics and comorbidities of the study population at baseline.

	Active	Placebo	*p*
Gender (m/w)	22/22	26/20	0.473
Age (years)	64.0 ± 7.8	61.7 ± 8.3	0.248
Height (cm)	168 ± 11.7	169 ± 8.4	0.560
Body Weight (kg)	88.9 ± 17.7	92.3 ± 21.7	0.471
Diabetes Duration (years)	14.0 ± 8.8	11.6 ± 5.9	0.167
Metformin and Other OADs	24 (54.5)	27 (58.7)	0.663
OADs and Basal Insulin	20 (45.5)	19 (41.3)	0.670
Smoking (%)	9 (20.4)	12 (26.1)	0.210
Cardiovascular Disease *	12 (27.3)	15 (32.6)	0.490
Dyslipidemia	33 (75.0)	34 (73.9)	0.849
Hypertension	34 (77.3)	32 (69.6)	0.289
Metformin Therapy	12.9 ± 8.8	10.1 ± 5.8	0.125

Data are given as means ± standard deviation or *n* (%). OADs: oral antidiabetic drugs; see Section 2.3 * Cardiovascular disease was defined as myocardial infraction or stroke or atrial fibrillation or cardiac arrhythmias or cerebrovascular disease.

**Table 2 nutrients-13-00395-t002:** Laboratory measurements at baseline in both groups.

	Active	Placebo	*p*
HbA1c (%)	6.79 ± 0.7	6.82 ± 0.7	0.874
HbA1c (mmol/L)	50.7 ± 7.6	51.0 ± 7.9	0.873
Vitamin B12 (pmol/L) *	232.0 ± 71.8	230.9 ± 85.9	0.788
White Blood Count (10^3^/μL)	7.83 ± 1.3	8.13 ± 0.8	0.246
Hemoglobin (g/dL)	13.82 ± 1.4	13.2 ± 1.2	0.228
Mean Corposcular Volume (fL)	88.6 ± 5.4	87.5 ± 5.4	0.392
Platelets (10^3^/μL)	258.5 ± 35.1	247.8 ± 54.2	0.399
Folic acid (ng/mL)	16.4 ± 8.5	25.7 ± 13.6	0.172
Creatinine (mg/dL)	0.94 ± 0.3	0.85 ± 0.2	0.107
Cholesterol (mg/dL)	174.6 ± 43.8	175.5 ± 48.1	0.934
Triglycerides (mg/dL)	161.6 ± 50.1	148.9 ± 51.7	0.327
High Density Lipoprotein (mg/dL)	46.1 ± 3.1	44.8 ± 11.1	0.661
Low Density Lipoprotein (mg/dl)	104.1 ± 47.2	102.6 ± 45.8	0,580

**Abbreviations:** HbA1c: glycated hemoglobin; * To convert to pg/mL multiply by 1.355. See Appendix B, Table A3 for reference intervals.

**Table 3 nutrients-13-00395-t003:** Indices of cardiovascular autonomic reflex tests (CARTs) and neuropathy tests at baseline.

	Active	Placebo	*p*
MNSIQ	5.8 ± 2.2	5.9 ± 2.1	0.772
MNSIE	3.6 ± 2.4	3.8 ± 2.28	0.798
DQOL	39.9 ± 10.3	40.2 ± 11.1	0.722
SNAP (μV)	5.2 ± 4.3	5.1 ± 4.2	0.757
SNCV (m/s)	28.2 ± 22.7	34.8 ± 24.6	0.132
VPT (V)	31.5 ± 14.2	26.8 ± 13.7	0.098
MCR	9.2 ± 10.2	16.3 ± 25.0	0.113
Valsalva	1.51 ± 0.21	1.52 ± 0.3	0.633
PI	4.3 ± 5.8	2.9 ± 2.5	0.362
PO (mmHg)	8.3 ± 12.2	7.3 ± 9.5	0.880
Pain score	18.4 ± 9.2	19.3 ± 8.5	0.397
ESCF (μS)	72.8 ± 10.1	72.4 ± 12.3	0.901
ESCH (μS)	69.1 ± 12.5	69.2 ± 10.0	0.684

**Abbreviations**: MNSIQ: Michigan Neuropathy Screening Instrument Questionnaire; MNSIE: Michigan Neuropathy Screening Instrument Examination; DQoL: Diabetes Quality of Life Questionnaire; SNAP: sural nerve action potential (amplitude); SNCV: sural sensory nerve conduction velocity; VPT: vibration perception threshold; MCR: mean circular resultant; PI: postural index; postural hypotension: PO (orthostatic hypotension; Pain: pain score questionnaire; ESCF: electrochemical skin conductance in feet; ESCH: electrochemical skin conductance in hands.

**Table 4 nutrients-13-00395-t004:** Changes in indices from baseline to follow-up in both groups.

	Active			Placebo			
	Baseline	12 months	*p* ^a^	Baseline	12 months	*p* ^b^	*p* ^c^
HbA1c (%)	6.79 ± 0.7	6.72 ± 0.7	0.257	6.82 ± 0.7	6.73 ± 0.6	0.312	0.876
HbA1c (mmol/L)	50.7 ± 7.6	49.9 ± 7.6	0.257	51.0 ± 7.9	50.02 ± 6.5	0.312	0.876
B12 (pmol/L)	232 ± 71.8	776.7 ± 242.3	**<0.001**	230.9 ± 85.9	242.8 ± 100.7	0.338	**<0.001**
MNSIQ	5.8 ± 2.2	5.44 ± 2.1	**0.002**	5.97 ± 2.1	6.17 ± 2	**0.017**	**<0.001**
MNSIE	3.55 ± 2.4	3.59 ± 2.3	0.663	3.8 ± 2.8	3.6 ± 2.1	0.617	0.607
DQOL	39.9 ± 10.3	38.1 ± 9.5	**<0.001**	40.2 ± 11.1	40.2 ± 11.1	0.931	**0.001**
SNAP (μV)	5.2 ± 4.3	7.3 ± 4.7	**<0.001**	5.1 ± 4.2	4.6 ± 4	**<0.001**	**<0.001**
SNCV (m/s)	28.2 ± 22.7	30.31 ± 23.2	**<0.001**	34.8 ± 24.6	32.9 ± 24.2	**0.045**	**<0.001**
VPT (V)	31.5 ± 14.2	23.8 ± 13.6	**0.001**	26.8 ± 13.7	25.8 ± 13.4	0.250	**0.007**
MCR	9.2 ± 10.2	12.6 ± 20.4	0.409	16.3 ± 25	7.99 ± 11.8	**0.025**	0.452
Valsalva	1.51 ± 0.21	1.60 ± 0.23	0.983	1.52 ± 0.3	1.59 ± 0.24	0.853	0.761
PI	4.3 ± 6.4	4.5 ± 6.7	0.700	2.9 ± 2.6	3.2 ± 9.4	0.875	0.954
PO (mmHg)	8.3 ± 12.2	5.2 ± 9.7	0.115	7.3 ± 9.5	7.8 ± 9	0.679	0.108
Pain Score	18.4 ± 9.7	17.1 ± 9	**<0.001**	19.3 ± 8.5	20.9 ± 8.5	**<0.001**	**<0.001**
ESCF (μS)	72.8 ± 10.1	74.5 ± 10.1	**0.014**	72.4 ± 12.3	71.2 ± 11.6	0.142	**0.008**
ESCH (μS)	69.14 ± 12.5	71.07 ± 12.4	0.260	69.2 ± 10	69.3 ± 7.8	0.636	0.265

^a^ For difference during follow-up in the active group; ^b^ for difference during follow-up in the placebo group; ^c^ for difference between groups adjusted for HbA1c and antidiabetic medication. Abbreviations: HbA1c: glycated hemoglobin; MNSIQ: Michigan Neuropathy Screening Instrument Questionnaire; MNSIE: Michigan Neuropathy Screening Instrument Examination; DQoL: Diabetes Quality of Life Questionnaire; SNAP: sural nerve action potential (amplitude); SNCV: sural sensory nerve conduction velocity; VPT: vibration perception threshold; MCR: mean circular resultant; PI: postural index; Postural hypotension: PO (orthostatic hypotension); Pain: pain score questionnaire; ESCF: electrochemical skin conductance in feet; ESCH: electrochemical skin conductance in hands.

## Data Availability

The data are available upon request.

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
