# Peer review of "Vitamin B12 Supplementation in Diabetic Neuropathy: A 1-Year, Randomized, Double-Blind, Placebo-Controlled Trial"

_nutrients, 2021, doi:10.3390/nu13020395_

Round 1

Reviewer 1 Report

The article by Didangelos et al. investigates in a placebo-controlled, parallel-arm design the effects of oral vitamin B12 supplementation over 12 months in subjects with diabetic autonomic and sensorimotor neuropathy. This is a well-done study and an excellently written manuscript. The therapy of diabetic sensorimotor neuropathy is largely a symptomatic one, which leaves the substrate of the disease, e.g. the nerve damage, untreated. There is an immense need for so-called pathogenetically oriented treatments that provide also nerve functional and morphological regeneration. Vitamin B12 might represent such a therapy and the results of the present study are of great importance since they show indeed an improvement not only in neuropathic symptoms, but also in functional parameters, suggesting nerve regeneration. The long duration of 1 year adds to the value of these data. Moreover, the study allows practical consequences e.g. clear definition of interventional groups (vitamin B12 levels < 400 pmol/L in long-term metformin-treated patients) and the fact that an oral supplementation of 1000 µg/day is enough to see clinical improvement.

Overall congratulation to the authors for the performance of this important study. Only some minor comments apply:

  • At study entry, how were other vitamin B12 absorption abnormalities (e.g. atrophic gastritis) ruled out?
  • Line 76-77: rephrase: “…that vitamin B12 levels should be regularly, yearly in patients on metformin 76 and neuropathy, monitored”
  • Line 138-139: GLP-1 analogues are not “oral” antidiabetic drugs.
  • Line 194-203: some of the information given here (e.g. “No patient received a sulfonylurea”) is redundant to the description given in chapter 2.3 or the tables. Please delete data and keep the reference.

Author Response

We appreciate the opportunity to submit a revised version of our manuscript ‘Vitamin B12 Supplementation in Diabetic Neuropathy: A 1-Year Randomized Double-Blind Placebo-Controlled Trial’.  We thank you and the reviewers for the very helpful comments to improve the quality of the manuscript. We have responded in detail to all comments and revised the manuscript accordingly. Changes in the text are marked in red with the “track changes function” of Microsoft Word.

We are pleased to hear the positive comments of the reviewer on our manuscript and thank him/her for the helpful suggestions to improve the quality of our work.

Minor concerns

Comment 1: “At study entry, how were other vitamin B12 absorption abnormalities (e.g. atrophic gastritis) ruled out?”

Response: As mentioned in the exclusion criteria, page 3 lines 117-123, we excluded from the study, patients with pernicious anemia, malabsorption syndromes or other causes of B12 deficiency. This was made on the basis of a free history,  absence of any clinical symptoms or signs and the absence of hematological abnormalities, particularly pernicious anemia, leukocytopenia or thrombocytopenia. Atrophic gastritis is expected to be accompanied by Biermer’s pernicious anemia. Therefore, we did not consider endoscopy to be necessary for ruling out atrophic gastritis in our patients.

Comment 2: “Line 76-77: rephrase: “…that vitamin B12 levels should be regularly, yearly in patients on metformin 76 and neuropathy, monitored”

Response: We  thank the reviewer for this comment. We have now rephrased the sentence, on page 2, line 76, which now reads:

“..that in patients with diabetic neuropathy on metformin therapy,  vitamin B12 levels should be regularly monitored in an annual basis

Comment 3: “Line 138-139: GLP-1 analogues are not “oral” antidiabetic drugs.”

Response: Certainly the reviewer is right. However, it is common practice for diabetologists to study GLP-1 analogues together with “true” oral antidiabetic drugs (OADs). This is done to contrast all these antidiabetic drugs to insulin therapy. We therefore prefer not to mention GLP-1 analogues separately from other OADs.

Comment 4: “Line 194-203: some of the information given here (e.g. “No patient received a sulfonylurea”) is redundant to the description given in chapter 2.3 or the tables. Please delete data and keep the reference”

Response: We thank  the reviewer for this comment. We agree with him and therefore have now deleted the redundant details from lines 194-203 and kept the reference in the legend of table 1.

Reviewer 2 Report

this manuscript reports the efficacy of vit B12 oral supplementation in Patients with T2DM and neuropathy. the main result is the improvement of neural damage and function. 

the conclusion should be expected : correction of low Vit B12 with oral supplementation improves neural function.

Authors should provide some explanation:

1) how vit B12 oral supplementation overcome metformin inhibition of its absorption?

2) can metformin suspension correct  low B12 after 12 month

3) commonly VitB12 is prescribed as injection: authors should discuss and compare results obtained with oral and i.m. injection

Author Response

We appreciate the opportunity to submit a revised version of our manuscript ‘Vitamin B12 Supplementation in Diabetic Neuropathy: A 1-Year Randomized Double-Blind Placebo-Controlled Trial’.  We thank you and the reviewers for the very helpful comments to improve the quality of the manuscript. We have responded in detail to all comments and revised the manuscript accordingly. Changes in the text are marked in red with the “track changes function” of Microsoft Word.

We would like to thank the reviewer for taking the time to perform this review and for his/her instructive comments.

Comment 1: “how vit B12 oral supplementation overcome metformin inhibition of its absorption?”

Response: As we mentioned on page 2, lines 80-90, metformin reduces Β12 absorption from the terminal ileum in about 30% of patients and in a dose- and duration- dependent manner (line 85). However, the exact mechanisms by which chronic treatment with metformin results to vitamin B12 inadequacy are not clear yet. It has been shown the administration of large doses of vitamin B12  given orally leads to a significant increase of B12 levels (please see reference #66 and #67, as well as Vidal-Alaball J et al, Cochrane Database Syst Rev 2005 Jul 20;(3)).  It may be that with larger doses, a larger amount of B12 is absorbed which is enough for increasing B12 levels. Furthermore, certain amounts of vitamin B12 may be absorbed directly from the mouth (please see reference #66). Indeed, in our previous study (reference #51) oral administration of 250 mcg of vitamin B12 significantly increased B12 levels.

Comment 2: “can metformin suspension correct  low B12 after 12 months”

Response: Obviously, metformin suspension would lead to an increase of B12 levels, but whether B12 levels would be normalized and in which time frame (12 months, earlier or later) cannot be known in advance. After all, metformin is considered to be the first-line treatment of choice for almost every patient with DM2, because of its few side effects and very good long-term effects in the course of DM2. Therefore, suspension of metformin would not be reasonable for most DM2 patients.

Comment 3: “commonly VitB12 is prescribed as injection: authors should discuss and compare results obtained with oral and i.m. injection”.

Response: We agree with the reviewer that previously vitamin B12 was commonly administered intramuscularly in order to overcome intrinsic factor deficiency. However, only 10% of the injected dose is absorbed (Bensky et al, 2019). Recent literature points out that vitamin B12 can also be given orally, sublingually or intranasal. As also mentioned on page 8, line 345-348, oral administration provides patient autonomy, increased patient satisfaction, reduced treatment costs and fewer risks for patients receiving anti-coagulants. Oral administration has been proven to be at least equally effective compared to i.m. injections (Bensky et al, 2019 ;Parry-Strong et al, 2016) and to produce equal hematological and neurological responses (Vidal-Alaball et al, Cochrane Database Syst Rev 2005 Jul 20;(3)). High oral doses (1000-2000 μg daily) are to 0.5-4% absorbed by passive diffusion, a mechanism that does not require the presence of intrinsic factor or a functioning terminal ileum. Moreover, the oral dispersible tablet we used has the additional advantages of allowing vitamin B12 absorption by sublingual capillaries (see reference #66 and #67), thus being well accepted by patients with swallowing difficulties and ensuring a higher adherence rate. We have now added the relevant text on page 8, lines 355-358.

Round 2

Reviewer 2 Report

I did not find any comments on the limitation of the experimental design

Author Response

Several limitations of our study were already mentioned in lines 402-406 of the previous version of our manuscript. Upon the request of the reviewer, we have now added more limitations and more details on them on   page 9, lines 412-430, which now read:.

“Our study has certain limitations. All participants were followed in a single diabetes center and are Caucasians. This ensures homogeneity of the sample population, but does not allow for extrapolation of the results to other populations. We did not measure Glutamic Acid Decarboxylase, intrinsic factor or parietal cells autoantibodies, because our population had confirmed DM2 and we had excluded patients with pernicious anemia. In addition, we measured the functionality of sural nerve only and not of nerves in arms and hands. Sural nerve function assessment is widely accepted to represent the respective function of other nerves too. We also did not measure homocysteine and methylmalonic acid, because they are often not available in the daily practice of the diabetes clinic, have high costs and are not necessarily needed for the diagnosis of vitamin B12 deficiency.”